# Effects of Drying Treatments on the Physicochemical Characteristics and Antioxidant Properties of the Edible Wild Mushroom *Cyttaria espinosae* Lloyd (Digüeñe Mushroom)

**DOI:** 10.3390/jof11010013

**Published:** 2024-12-28

**Authors:** Marcelo Villalobos-Pezos, Ociel Muñoz Fariña, Kong Shun Ah-Hen, María-Fernanda Garrido Figueroa, Olga García Figueroa, Alexandra González Esparza, Luisbel González Pérez de Medina, José Miguel Bastías Montes

**Affiliations:** 1Faculty of Agricultural and Food Sciences, Graduate School, Austral University of Chile (UACh), Valdivia 5090000, Chile; mvillalobospezos@gmail.com; 2Institute of Food Sciences and Technology, Austral University of Chile (UACh), Valdivia 5090000, Chile; kshun@uach.cl (K.S.A.-H.); maria.garrido@alumnos.uach.cl (M.-F.G.F.); olga.garcia@uach.cl (O.G.F.); alexandra.gonzalez@uach.cl (A.G.E.); 3Laboratory of Biomaterials, Department of Chemical Engineering, Faculty of Engineering, University of Concepción (UdeC), Concepción 4030000, Chile; luisbgonzalez@udec.cl; 4Department of Food Engineering, University of Bío-Bío, Chillán 3780000, Chile; jobastias@ubiobio.cl

**Keywords:** edible wild mushrooms, drying treatments, ultrasonic-assisted extraction, phenolic compounds, antioxidant activity, ergothioneine, *C. espinosae* Lloyd

## Abstract

The wild mushroom *Cyttaria espinosae*, also known as digüeñe, is a parasitic ascomycete of *Nothofagus* trees endemic to southern Chile. This species of wild mushroom is of great nutritional importance, especially for the Mapuche indigenous communities, and is highly sought after. Edible wild mushrooms, rich in bioactive compounds, are a potential source of health-promoting components. In the case of *C. espinosae*, research on its bioactive compounds is still lacking as is research, on the effect of preservation treatments on these compounds due to its perishability. This study evaluates the effects of three drying treatments; freeze-drying, hot-air drying, and microwave–vacuum drying. The rehydration capacity, color, and microstructural properties of dried mushrooms were evaluated using SEM, while, total phenolic content, antioxidant activity determined by DPPH and ORAC assays, and ergothioneine levels were investigated in both fresh and dried extracts of *C. espinosae*. The results showed that freeze-drying and microwave–vacuum drying are recommended treatments for the digüeñe mushroom due to superior outcomes in rehydration rate, color, and structural properties observed through SEM images. Total phenolic content and antioxidant activity were higher in mushroom extracts dried by microwave–vacuum compared to the other drying treatments.

## 1. Introduction

In Chile, there are around 53 species of edible wild mushrooms, and in the Valdivian temperate forests, it is possible to collect around 30 species, among which several species of the genus *Cyttaria* stand out. This genus has coevolved specifically with different species of *Nothofagus* [1]. The *Nothofagus* forests harbor a great diversity of fungi, with a high degree of endemism, including ectomycorrhizal, saprophytic, and parasitic species [2]. Mushrooms of the genus *Cyttaria*, are parasitic on species of the genus *Nothofagus*, producing fruiting bodies in clusters on branches during September and October, signalling the beginning of spring [1,3]. Specifically, the *Cyttaria espinosae* mushroom fruits on *Nothofagus obliqua*, *Nothofagus glauca*, and *Nothofagus procera*. It is very similar in appearance to *Cyttaria hariotii* [3], a well-studied mushroom of the genus *Cyttaria* [4,5]. This edible wild mushroom is highly sought after; it can be eaten raw without any seasoning or in a salad dressed with salt, chili, and coriander [3]. Wild mushroom species are of great importance nutritionally, especially for the indigenous Mapuche communities [6], and *C. espinosae*, known commonly as digüeñe, quireñe, or lihueñe, has continued to be collected over time, with a drastic increase in recent years to become the most consumed digüeñe species throughout its geographical distribution in Chile [7]. Its occurrence appears to have declined, mainly due to habitat loss, modification, and/or fragmentation [6]. Despite the seasonal nature of edible wild mushrooms, the inclusion of such foods in the diet, even in small or sporadic amounts, promotes an enriched diet and plays an important role in food sovereignty [2]. In many European countries, the consumption of wild mushrooms is preferred to cultivated species, and more recently, foraging has become a highly valued leisure activity, both as part of the cultural heritage and as an economically profitable activity for portions of the rural population [8].

Edible mushrooms research has included studies of their chemical composition and nutritional value; understanding the chemical composition of wild mushrooms allows them to be used as food or bioactive resources, benefiting local economies related to foraging, gastronomy, and health, while promoting better management and conservation of the habitats where they are collected [4]. However, knowledge of the nutritional value of wild mushrooms remains limited. Historical traditions and extensive research in East Asian countries have demonstrated the preventive and therapeutic properties of many mushroom species, with recent research highlighting interest in antioxidant constituents and various phenolic compounds, which appear to be the most potent group of antioxidants [9]. Phenolic compounds are among the most potent and therapeutically useful bioactive substances, providing health benefits associated with reduced risk of chronic and degenerative diseases [10]. For both *C. espinosae* and other species within the genus *Cyttaria*, research is still lacking on their nutritional value and bioactive compounds, such as phenolic compounds with antioxidant, anti-inflammatory, or anti-tumor properties [7].

The drying of food is an ancient method of preservation in human history; its purpose is to reduce microbial growth by lowering the moisture content of the food, thereby increasing its shelf life [11]. Several methods have been developed. Freeze-drying is a technique that produces high quality dried foods and compared to thermal treatment processes, freeze-drying results in low nutrient degradation. However, it is an expensive method [12]. Microwave–vacuumdrying is another method with advantages including rapid energy transfer during microwave drying and rapid mass transfer at low temperatures during vacuum heating [13]. This drying method shortens the processing time, uses a lower moisture reduction temperature and preserves the antioxidant properties of the food, as the heat penetrates more quickly throughout the food during microwave–vacuum drying. In contrast, during traditional thermal drying processes, such as hot-air drying, heat is transferred more slowly from the outer layer to the core of the food [14].

The demand for new functional ingredients or bioactive compounds of natural origin is steadily increasing, leading in recent years to a growing interest in the extraction of bioactive compounds from foods such as mushrooms [15]. Bioactive compounds from mushrooms are valuable resources and extraction methods are critical [16]. Ultrasound-assisted extraction is effective due to the acoustic cavitation effect generated in the solvent by the passage of ultrasonic waves, which can cause cell disruption and increase the surface area of contact between the solid and liquid phases, thereby increasing the yield of secondary metabolites [17]. Extraction yield is influenced by other variables such as solvent/solid ratio, solvent composition and concentration, contact time, and extraction temperature. Therefore, it is important to define conditions that maximize the extraction of target compounds for each matrix [18]. Ultrasound-assisted extraction is also considered to be the most promising technology for the extraction of biologically active compounds [19]. Due to the close relationship between phenolic compounds and antioxidant activity [20], mushroom extracts are becoming increasingly popular as antioxidants [21].

The aim of this research was to determine the conditions of the variables: temperature, time, and solvent, to maximize the extraction of total soluble phenolic content from *C. espinosae* through a response surface model (RSM). In addition, the drying treatments were compared in terms of color parameters, microstructural properties, and rehydration rates of dried *C. espinosae* mushrooms, as well as total soluble phenolic content, antioxidant activity, and ergothioneine content of fresh and dried mushroom extracts.

## 2. Materials and Methods

### 2.1. Samples of the Edible Wild Mushroom C. espinosae

The fruiting bodies of the *Cyttaria espinosae* mushroom (Digüeñe) were collected from fields of the Nothofagus Forest in the mountainous regions of the cities Chillán and Valdivia, Chile, during the 2022 and 2023 harvest seasons. They were stored at −80 °C until analysis.

### 2.2. Drying Processes

Before drying, the *C. espinosae* mushrooms were removed from the freezer and allowed to equilibrate to room temperature before each treatment. They were selected according to size and homogenized. The mushrooms were halved and dried by three different methods: Freeze-drying (FD), hot-air drying (HAD), and microwave–vacuum drying (MVD). For each drying method, samples of 20 ± 1 g of fungi were used and experimental assay was performed in triplicate. The drying process was carried out until a constant weight was reached. The initial moisture content of fresh *C. espinosae* mushrooms was 89%. The final moisture content of the fungi dried by freeze-drying, hot-air, and microwave–vacuum methods was 7.36 ± 0.10%, 10.52 ± 0.20%, and 9.02 ± 0.32%, respectively. Tests were performed in triplicate.

#### 2.2.1. Freeze-Drying

Freeze-drying was conducted as described by Tamarit-Pino et al. [22]. The samples were dried in a freeze dryer (Beta 1–8 LDplus, Christ, Osterode am Harz, Germany) at a pressure of 0.021 mbar and a condenser temperature of −55 °C until constant weight was reached. The weight loss in the freeze-drying process was determined using a weighing system installed in the vacuum chamber monitored with an Arduino Uno microcontroller and coupled between the freeze-dryer and a computer.

#### 2.2.2. Hot-Air Drying

The hot-air drying was carried out as described by Giri & Prasad [13], with some modifications. The samples were dried in an air flow oven (ZRD-A5055, Zhicheng, China) at temperatures of 40, 60, and 80 °C. To determine the drying kinetics, samples were weighed every 10 min using a digital electronic scale until a constant weight was achieved.

#### 2.2.3. Microwave–Vacuum Drying

The microwave–vacuum drying was carried out as described by Kantrong, et al. [23], with modifications. The samples were dried in a microwave–vacuum oven (ZYWX—1 KW, Zhengshou Keda Machinery and Instrument Equipment Co., Zhengzhou, Henan, China) at microwave powers of 160, 210, and 260 W at an absolute pressure of 20 kPa (160 W/20 kPa, 210 W/20 kPa, and 260 W/20 kPa). The samples were weighed every 2 min using a digital electronic scale to determine the drying kinetics until constant weight was achieved.

### 2.3. Color Measurement

The color measurement of fresh and dried *C. espinosae* samples was determined as described by Hou et al. [24]. The color parameters of the samples were evaluated by reflectance with a colorimeter (CR-400, Minolta Chroma Meter, Tokyo, Japan) that was previously calibrated with a blank. The results were expressed as CIELAB values. In the CIELAB coordinates, the values L*, a*, and b* describe a three-dimensional color space. The L* value is the vertical axis and defines lightness, while the a* and b* values are perpendicular horizontal axes and define the range from red to green and from blue to yellow, respectively [25], and ΔE values = (ΔL*^2^ + Δa*^2^ + Δb*^2^)^½^, which represent the colorimetric difference between the sample and the standard white reflectance plate [26]. The assay was performed in triplicate.

### 2.4. Rehydration Rate of Dried C. espinosae

The rehydration rate (Rf) was carried out according to Qiu et al., [27], with some modifications. During the rehydration process, samples were removed from the water bath at intervals of 30 s up to the first minute, then every 2 min up to 10 min, then every 5 min up to 30 min, then every 10 min up to 60 min, and finally every 30 min up to 300 min until a constant weight was reached. The samples were weighed after draining on filter paper using a digital electronic balance. Rf was calculated according to the following formula Rf = (Gf/Gg)*100, where Gf is the weight of the rehydrated and drained sample and Gg is the weight of the original dry sample. The procedures were carried out in triplicate.

The swelling mechanisms were determined using the Peppas models, and the nature of the diffusion mechanism was analyzed by fitting the data to the models presented in Table 1.

### 2.5. Microstructure of Dried C. espinosae

A scanning electron microscope (SEM) (EVO, Zeiss, German Carl Zeiss, EVO, Cambridge, UK) was used to visualize the microstructure of the stipe (stem, peduncle) of the dried mushrooms. Selected and cut samples were mounted on carbon adhesive tape on slides, adding bridges of the same material to improve electrical conductivity, and were coated with gold/palladium using a sputter coater (Leica Microsystems GmbH, EM ACE200, Wetzlar, Germany).

### 2.6. Ultrasound-Assisted Water Bath Extraction of C. espinosae

The extraction method was performed according to Leiva-Portilla et al. [31], with some modifications. A total of 2 g of raw mushrooms were weighed to optimize the extraction of the total soluble phenolic content, 4 g of raw mushrooms, and 0.4 g of dried mushrooms were weighed to compare the effects of the drying treatment. The raw mushrooms were ground and homogenized in an immersion blender (2616, Oster, China) and the dried mushrooms were pulverized in a coffee grinder (Sindelen, MOL—165, La Florida, Santiago, Chile). They were weighed into a 50 mL centrifuge tube. Then 15 mL of distilled water solvent was added, and they were manually homogenized and then extracted in an ultrasonic water bath at 50 °C (Neytech, Ultrasonik, 57H, Yucaipa, CA, USA). At the end of each extraction time, the 50 mL tubes were centrifuged at 3000× *g* for 10 min (Rotofix 32A, Hettich, Tuttlingen, Germany), the supernatant was transferred to a 50 mL centrifuge tube and the precipitate was resuspended with the solvent, thus completing three extraction cycles.

### 2.7. Optimization of the Extraction Procedure Using Response Surface Methodology for Fresh C. espinosae Extracts

Response surface methodology was applied to investigate the effect of three independent variables, the ratio of distilled water/ethanol (100/0, 50/50, and 0/100% *v*/*v*), extraction time (30, 60, and 90 min), and extraction temperature (50, 65, and 80 °C) on the dependent variable, total phenolic content in *C. espinosae* extracts. The experiments were conducted using a 3-level factorial design in three blocks, utilizing Statgraphics Plus v.5.1 software. The complete design consisted of 30 experimental runs, including one central point per block, and these were performed in triplicate.

### 2.8. Determination of Total Soluble Phenolic Content in Fresh and Dried Edible Wild Mushroom Extracts

Total soluble phenolic content of the extracts was measured by the Folin–Ciocalteu method [32]. A volume of 40 µL of extract sample was diluted in 3160 µL of distilled water and mixed with 200 µL of Folin–Ciocalteu reagent. The mixture was vortexed for 7–10 s, then 600 µL of 20% sodium carbonate (Na_2_CO_3_·5H_2_O) was added, and it was vortexed again for 7–10 s. The reaction developed in the dark, protected from sunlight, for 120 min at room temperature. Absorbance was measured at 765 nm in a spectrophotometer (Spectronic, Genesys 5, Rochester, NY, USA). Quantification was performed against a gallic acid calibration curve. The results of total phenolic content were expressed as mg of gallic acid equivalent per gram of dry mass (mg GAE/g of d.m.). All measurements were performed in triplicate.

### 2.9. Determination of Antioxidant Activity of Fresh and Dried Edible Wild Mushroom Extracts

#### 2.9.1. DPPH Radical Scavenging Activity

The radical scavenging activity of the extracts from the samples was determined through the 2,2-diphenyl-1-picrylhydrazyl (DPPH) assay according to the method of Brand et al. [33]. From an aliquot of the sample extract previously centrifuged at 10,000× *g* for 10 min, 100 µL was added to 2.9 mL of 1 mM DPPH solution in methanol in a glass cuvette and mixed completely for 10 s. The absorbance of the samples was measured at 515 nm using a spectrophotometer (Spectronic, Genesys 5, Rochester, NY, USA). Results were expressed as mmol Trolox equivalent per 100 g of dry mass (mmol TE/100 g d.m.). The assay was conducted in triplicate.

#### 2.9.2. Oxygen Radical Absorption Capacity (ORAC) Assay

The capacity of the extracts to delay the oxidative decomposition of fluorescein induced by 2,2′-azobis (2-methylpropionamidine) dihydrochloride (AAPH) was measured as described by Ou et al. [34]. A previously prepared aliquot of the sample extract was centrifuged at 10,000× *g* for 10 min. Then, these samples of extract and reagents were prepared and diluted in 75 mM phosphate buffer (pH 7.4) in a black microplate with a flat bottom and transparent. Volumes of 45 μL of the sample and 175 μL of fluorescein at 108 nM were deposited. This mixture was preincubated for 30 min at 37 °C; after this time, 50 μL of AAPH solution at 108 nM was added. The microplate was immediately placed in the dual-scanning microplate spectrofluorometer (Molecular Devices, Gemini XPS, Sunnyvale, CA, USA) for 60 min using a 538 nm emission and 485 nm excitation wavelength. The fluorescence readings were recorded every 3 min, and the microplate was automatically shaken before and after each reading.

The microplate was immediately placed in the double-scanning microplate spectrofluorometer for 60 min with excitation and emission wavelengths set at 485 nm and 538 nm, respectively. Fluorescence readings were recorded every 3 min. The microplate was automatically shaken before and after each reading. Trolox at 6, 12, 18, and 24 μM as a standard to obtain the calibration curve and the buffer was used as a blank. ORAC values were calculated as area under the curve (AUC) values and compared with the Trolox calibration curve. Results were expressed as mmoles equivalent of Trolox per 100 g of dry mass (mmoles TE/100 d.m.). The assay was conducted in triplicate.

### 2.10. Ergothioneine Content of Fresh and Dried Edible Wild Mushroom Extracts

The determination of the ergothioneine content in the extracts was performed by high-performance liquid chromatography (HPLC), in accordance with Dubost et al. [35]. Briefly, the chromatographic separation of ergothioneine was achieved using two reversed-phase columns (C18, 5 μm, 250 mm × 4.6 mm, InertSustain, GL Sciences, Tokyo, Japan) connected in tandem. The mobile phase was prepared as an isocratic solution consisting of 50 mM sodium phosphate buffer, 3% acetonitrile, and 0.1% trimethylamine, adjusted to pH 7.3, with a flow rate of 1 mL/min a wavelength of 254 nm was used for monitoring. The injection volume was 20 μL, with a column temperature of 30 °C. The identification and quantification of ergothioneine were confirmed by comparing retention times and areas of peaks from the sample extracts with those obtained from different concentrations of standard ergothioneine. Results were expressed as mg of ergothioneine per gram of dry mass (mg ergothioneine/g d.m.). The assay was conducted in triplicate.

### 2.11. Statistical Analysis

All assays were performed in triplicate. Results were reported as mean ± standard deviation. Analysis of variance and significant differences were evaluated using Simple ANOVA with Statgraphics Plus v. 5.1 software (Statistical Graphics Corp., Herndon, VA, USA). The Tukey multiple range test included in the statistical program was also used to demonstrate the existence of homogeneous groups within each of the parameters at a 95% confidence level. Data were plotted using OriginPro 10.1.5.132^®^ software (OriginLab Corporation, Northampton, MA, USA).

## 3. Results and Discussion

### 3.1. C. espinosae Mushoom Drying Processes

The variation of moisture ratio as a function of time during freeze-drying, hot-air drying, and microwave–vacuum drying of the mushrooms *C. espinosae* are shown in Figure 1a,b. Drying is often used in the processing of edible and highly perishable mushrooms to allow preservation and availability throughout the year [36]. Freeze-drying achieved equilibrium moisture after 1980 min of drying time, compared to 1590 min for hot-air drying at 40 °C. For hot-air drying, as expected, a decrease of 37.74% in drying time was observed with an increase in temperature from 40 to 60 °C (580 min) and of 61.62% in the 60 to 80 °C temperature range (380 min). For microwave–vacuum drying (Figure 1b), the drying time decreased as a function of the microwave power used [13]. Under a vacuum pressure of 20 kPa for the three microwave powers, a shorter drying time was achieved at 260 W/20 kPa (40 min), as expected for dried mushrooms [23], with a 16.6% reduction in drying time compared to at a microwave power of 160 W/20 kPa (48 min). Comparing the drying times of hot-air and microwave–vacuum drying, there was a reduction in microwave–vacuum drying time of 93.1%, slightly higher than reported in previous studies [37]. Figure 2 shows macroscopic images of *C. espinosae* mushrooms in the *Nothofagus obliqua* forests (Figure 2a), and before (Figure 2b) and after (Figure 2c) the application of the different drying treatments.

### 3.2. Color Measurement of Fresh and Dried Edible Wild Mushrooms

The most common methods of measuring color involve instruments that measure the reflectance of the surface. In the CIELab coordinates, the values L*, a*, and b* describe a three-dimensional color space. Color is an important parameter influencing consumer acceptability [23]. Drying processes had a significant effect on the color parameters of *C. espinosae*. Table 2 shows the values of L*, a*, b* and ΔE values. The values of L*, a*, b*, and ΔE for *C. espinosae* were 22.50, 6.01, 15.58, and 28.02, respectively. In general, as observed in this study, the L* value decreases after drying treatment compared to fresh mushrooms [38,39]. The L* value is used as an indicator to quantify the degree of browning of dehydrated foods [14]; and products with a high L* value are usually preferred by consumers [39]. The samples after freeze-drying had the highest L* value compared to other drying treatments for edible mushrooms [40]. *C. espinosae* mushrooms dried by microwave–vacuum showed relatively high L* values. No significant differences were observed between the microwave powers selected. The L* value of hot-air drying decreases with increasing temperature, a trend also observed for other edible wild mushrooms [41].

Regarding the a* and b* values (Table 2), they increase with the increase in temperature or microwave power [39], but this trend was not observed for the *C. espinosae* mushroom, since the a* and b* values for mushrooms dried with hot air at 60 °C had a significantly higher a* value compared to the a* values of mushrooms dried with hot-air at 40 °C and 80 °C. For the same hot-air drying condition at 60 °C, the b* value was higher compared to the other temperature conditions, but only significantly different from mushrooms dried at 80 °C. The same effect occurred for the a* and b* values of microwave–vacuum dried mushrooms, with no observed trend of decreasing values with increasing microwave power. Overall, higher a* and b* values were observed for microwave–vacuum drying with slightly significant differences between microwave–vacuum dried mushrooms and fresh *C. espinosae*. It is noteworthy that the microwave power of 210 W at a vacuum pressure of 20 kPa maintained a* and b* values similar to those of fresh mushrooms, with no significant differences observed. The significantly higher a* and b* values observed in the drying treatments other than freeze-drying compared to freeze-drying have already been reported in other edible mushrooms [40] and in *C. espinosae* [31].

The color difference ΔE indicates the degree of overall color change compared to the color of fresh mushrooms, and a well-dried mushroom should have a minimal difference in ΔE [23]. The ΔE values shown in Table 2 are listed as follows: Fresh > FD > MVD 210 W > MVD 260 W > MVD 160 W > HAD 60 °C > HAD 40 °C > HAD 80 °C. The smallest differences in ΔE compared to fresh mushrooms were those of freeze-dried mushrooms, followed by those of microwave–vacuum dried mushrooms. Significant differences were observed between these drying treatments. Slightly significant differences were observed for microwave–vacuum dried mushrooms at 210 W/20 kPa compared to freeze-dried and fresh mushrooms. For hot-air dried mushrooms, a greater difference in ΔE value was observed with increasing drying temperature, especially when the drying temperature increased significantly from 60 to 80 °C. This trend, observed in the differences among L*, a*, b*, and ΔE values between fresh *C. espinosae* mushrooms and those dried by hot-air and microwave-vacuum, has been previously observed in edible mushrooms [38].

### 3.3. Microstructure of Fresh and Dried Edible Wild Mushrooms

The influence on the quality of dried mushrooms was studied using scanning electron microscopy (SEM). SEM images of internal and surface structures of freeze-dried *C. espinosae* mushrooms are shown in Figure 3, while hot-air dried mushrooms are depicted in Figure 4. The SEM images of microwave–vacuum dried mushrooms are presented in Figure 5. The drying processes studied have very different influences on the microstructural properties of the *C. espinosae* mushroom. Freeze-dried mushrooms exhibit a more porous structure, both internally (Figure 3a,b) and on the surface (Figure 3c,d). These attributes, resulting from the low drying temperature, contribute to the high-quality product characteristic of this process [42]. As regards the internal structure of the hot-air dried mushrooms, porosity and alveoli are preserved in mushrooms dried at 60 °C (Figure 4e,f), compared to mushrooms dried at 40 °C (Figure 4a,b) and 80 °C (Figure 4i,j), where greater shrinkage and contraction of the alveoli are observed. On the surface, in the same drying process, porosity can be observed in descending order in fungi dried at 80 °C (Figure 4k,l) > 60 °C (Figure 4g,h) > 40 °C (Figure 4c,d), with greater overlapping of surface structures and more contraction in the latter drying condition (Figure 4c,d). Microwave–vacuum dried mushrooms (Figure 5a–l) show greater porosity in both internal structure and surface, with less shrinkage and contraction. These findings are consistent with previous research on edible mushrooms, where greater porosity and open structure is observed in microwave–vacuum dried mushrooms compared to hot-air dried mushrooms [13,38], and compared to other drying processes [38]. In microwave–vacuum dried mushrooms, greater porosity and pore size can be observed with increasing microwave power ratios and vacuum pressure [23]. The largest pore size can be observed in the mushroom dried at 210 W/20 kPa (Figure 5e,f) compared to mushrooms dried at 160 W/20 kPa (Figure 5a,b). For mushrooms dried at 260 W/20 kPa (Figure 5i,j), a larger pore size is observed compared to the other two microwave powers selected, but a collapse of the structure is observed in the loss of continuity in the porous structure, possibly due to the rate of moisture transfer from the food into the vacuum chamber [23]. This is consistent across the respective surfaces, which show greater porosity and less overlapping and contraction in mushrooms dried at 210 W/20 kPa (Figure 5g,h) compared to 160 W/20 kPa (Figure 5c,d) and especially at 260 W/20 kPa (Figure 5k,l).

### 3.4. Rehydration Rate

Dried mushrooms can be rehydrated by immersion in water prior to consumption. Rehydration characteristics are used as a quality parameter and indicate whether physical and chemical changes have occurred during drying due to process conditions [43]. Figure 6 shows the rehydration capacity of dried *C. espinosae* mushrooms for the different processes. The rehydration rate is listed in descending order as follows: FD > MVD 210 W > MVD 160 W > MVD 260 W > HAD 60 °C > HAD 80 °C > HAD 40 °C. Freeze-dried mushrooms have a higher rehydration capacity compared to those dried by microwave–vacuum and hot-air drying, and an even higher rehydration capacity than other drying methods for edible mushrooms [36,44], which can be explained by the structural differences mentioned above. In hot-air dried *C. espinosae* mushrooms, the increase in temperature did not have a negative effect on the rehydration capacity, contrary to what has been observed in other edible mushrooms [43]. It was found that the mushrooms dried at 40 °C had a lower rehydration rate compared to the other drying processes, supporting the observations made in the electron microscopy images that the rehydration rate was negatively affected. In general, hot-air drying processes have significant disadvantages in terms of rehydration properties [23], and the results of this research confirm this. Mushrooms dried by hot air have a lower rehydration rate than those dried by microwave-vacuum. For microwave–vacuum dried edible mushrooms, the rehydration rate is higher compared to hot-air dried samples [13,38], with the rehydration rate depending primarily on the vacuum pressure [13], which could lead to a more porous structure and a greater pressure difference between the vacuum chamber and the internal pressure of the food, potentially resulting in a less dense, more expanded, and fluffier structure, thus achieving a higher water absorption capacity [23]. A vacuum pressure of 20 kPa was evaluated in this research, similar to previous studies of edible mushrooms [38,45], and it was observed that microwave–vacuum drying at 210 W/20 kPa had a greater rehydration capacity compared to other microwave powers, suggesting that the ratio between microwave power and vacuum pressure is suitable for *C. espinosae* due to better preservation of structure and porosity [13,23]. The collapse of the structure in mushrooms dried at 260 W/20 kPa (Figure 5i,j) negatively affected the rehydration rate.

The mechanisms involved in the rehydration capacity were studied using the Peppas model (Table 3). For all the fungi analyzed, values of n were found to be less than 0.5, indicating that rehydration is determined by the rate of penetration of water molecules, suggesting that this process is governed by Fick’s diffusion law. The results show that the freeze-drying process has the highest rehydration rates (k). In hot-air drying treatments at different temperatures and microwave–vacuum drying, significant variations in the values of k and n were observed. In particular, the hot-air dried mushrooms at 40 °C show a considerably lower k value compared to the other treatments, suggesting that this process is slower and may be influenced by restrictive conditions. As the hot-air drying temperature increases, at 60 °C and 80 °C, the k value tends to increase, indicating more efficient rehydration. When comparing hot-air and microwave–vacuum drying methods, higher k values are observed for microwave-vacuum, demonstrating a better rehydration rate for fungi dried by this method. There is also a direct relationship among k values and microwave power ratio and vacuum pressure, with the highest rehydration rates achieved at 260 W/20 kPa. Data analysis shows that one-dimensional and three-dimensional diffusion models have higher R² coefficients compared to other models, suggesting that they are more representative of the rehydration process.

### 3.5. Effect of Different Extraction Factors in Ultrasonic-Assisted Water Bath on the Total Soluble Phenolic Content of C. espinosae Extracts

Temperature and time were controlled during the extraction procedures. The temperature of the water bath was maintained as constant as possible throughout the extraction process. The process variables were set at 100/0, 50/50, and 0/100% (*v*/*v*) distilled water/ethanol, extraction times of 30, 60 and 90 min and temperatures of 50, 65, and 80 ± 2 °C for the ultrasonic bath. The total soluble phenolic content of fresh *C. espinosae* mushrooms under the different combinations of the variables studied can be observed in Table 4, and the variables and their influence on the extraction of soluble phenolic content are presented in Table 5. The extraction of soluble phenolic compounds was significantly influenced by the factors temperature (X_1_) and solvent (X_3_) (*p* < 0.05). The interaction X_1_X_3_ was also significant (*p* < 0.05) (Table 5). The ultrasound-assisted extraction method could ensure the maximization of the extraction due to the mechanical effects that allow a greater penetration, improving the solvent transfer within the matrix cells, and the cavitation effect that causes the rupture of the cell walls and the release of their contents into the solvent, obtaining higher yields in less time at lower processing temperatures [46]. These results could therefore contribute to understanding of extraction methods for the soluble phenolic content of the digüeñe mushroom.

### 3.6. Response Surface Methodology

Figure 7 shows the interactions between temperature (X_1_), time (X_2_), and solvent (X_3_) and their mutual effect on the extraction of total soluble phenolic content (Y) from fresh *C. espinosae*. Figure 7a shows the interaction between temperature (X_1_) and time (X_2_). High values of soluble phenolic content are observed between 50 and 65 °C regardless of the time (X_2_), with minimum values of soluble phenolic content occurring when the temperature reaches 80 °C with an extraction time of 30 min. An increase in the value of the soluble phenolic content was observed when the extraction time reached 90 min, still below the values observed between 50 and 65 °C. Figure 7b illustrates the interaction between time (X_2_) and solvent (X_3_) and, regardless of the time, higher values of soluble phenolic content were observed when 100% distilled water was used in the distilled water/ethanol solvent. Figure 7c shows the interaction between temperature (X_1_) and solvent (X_3_), indicating that at 50 °C and with 100% distilled water in the solvent, the soluble phenolic content reached higher extraction values. The optimum conditions of temperature, time, and solvent for maximizing the extraction of soluble phenolic compounds from the *C. espinosae* were found to be 50 °C, 30 min, and 100% distilled water as solvent. Aqueous solvents generally provide a higher extraction yield compared to other solvents in wild mushrooms [47].

### 3.7. Determination of Total Soluble Phenolic Content in Aqueous Extracts of Fresh and Dried C. espinosae Mushrooms

In Figure 8, the total soluble phenolic content of fresh extracts of *C. espinosae* mushrooms and for each drying process can be observed. The drying processes lead to significant changes in phenolic compounds depending on the drying method. All antioxidants found in mushrooms are susceptible to various factors occurring during the drying process, and a higher total phenolic content has been observed in dried edible wild mushrooms compared to fresh and frozen mushrooms [36]. For *C. espinosae*, significant differences in total soluble phenolic content (Figure 8) were observed for extracts of mushrooms dried by microwave–vacuum at 210 W/20 kPa (9.13 mg GAE/g d.w.) and by freeze-drying (5.92 mg GAE/g d.w.). In descending order of total soluble phenolic content: MVD 210 W > MVD 260 W > HAD 80 °C > MVD 160 W > HAD 40 °C > HAD 60 °C > Fresh > FD. For hot-air dried samples, in contrast with what has been observed for other wild mushrooms dried with hot air, increasing drying temperature led to a decrease in soluble phenolic content [41]. The highest soluble phenolic content was obtained in extracts from *C. espinosae* mushrooms dried with hot air at 80 °C. This could be explained by the inactivation of polyphenol oxidase, which is thermally unstable above 60 °C [14], protecting thus from enzymatic degradation. The extraction of the soluble phenolic content in freeze-dried mushrooms by the Folin–Ciocalteu reaction was 5.74% lower than in fresh mushrooms, with slightly significant differences, which may have been influenced by the quality of the structure resulting from the drying treatment, since the extraction depends on the penetration of the solvent into the matrix and the diffusion of the phenolic compounds from the matrix, as well as the solutes from the matrix into the solvent [16]. The increase in phenolic content obtained in the extracts of mushrooms dried by both hot-air and microwave–vacuum suggests that the drying treatments could affect the structural changes in the cellular matrix of the mushrooms, as observed in SEM images (Figure 3, Figure 4 and Figure 5), allowing a greater release of phenolic compounds from the matrix compared to fresh mushrooms due to the weakening of the structure resulting from heat stress [48], and that the drying treatments could presumably favor enzymatic activation, which in turn promotes the production of secondary metabolites [49], resulting from the effect of drying temperature on the mushrooms, not only under drying conditions, but also under fruiting conditions [48]. The differences observed in the phenolic content of *C. espinosae* mushrooms dried by hot air and other drying treatments such as microwave and microwave–vacuum have been noted in previous research on edible mushrooms included in vitro assays [44,45]. This high retention of soluble phenolic content in the extracts of mushrooms dried by microwave–vacuum, especially at 210 W/20 kPa, was presumably due to the preserved structure of the cellular matrix (Figure 5e–h), resulting in shorter drying time, avoiding oxidation and loss of soluble phenolic content [45].

### 3.8. Antioxidant Activity and Ergothioneine of Aqueous Extracts from Fresh and Dried C. espinosae Mushrooms

Previous studies have shown that the antioxidant potential of foods is highly dependent on the phenolic content [50], which is also observed in edible mushrooms [51]. For the antioxidant activity of the DPPH assay (Figure 9a), significant differences are observed in the extracts of fresh mushrooms dried by freeze-drying and hot-air at 40 and 60 °C (0.63, 0.65, 0.81, 0.72 mmol TE/100 g dry weight, respectively) compared to the extract of mushrooms dried by hot air at 80 °C (1.73 mmol TE/100 g dry weight), with the highest antioxidant activity related to the extracts of fresh mushrooms and those subjected to other drying processes. It has been observed that the DPPH assay shows no correlation with phenolic content [52], which could be due to the presence of other compounds besides phenols that may also have DPPH radical scavenging activity [18]. For the antioxidant activity of the ORAC assay (Figure 9b), no significant differences were found between the aqueous extracts of fresh and dried mushrooms, although a trend was observed between phenolic content and antioxidant activity by the ORAC assay. The results of this research are consistent with previous studies on edible mushrooms, where a good correlation between ORAC values and phenolic content was found [9]. The trend of antioxidant activity observed in this research differs from a previous study on aqueous extracts of the mushroom *C. espinosae*, where significantly lower antioxidant activity was observed in the DPPH assay for aqueous extracts of mushrooms dried by hot air, and significantly higher antioxidant activity was observed in the ORAC assay for aqueous extracts of mushrooms dried by freeze-drying, compared to aqueous extracts of mushrooms dried by hot air, and even more so compared to extracts of fresh mushrooms [31]. These observed differences are probably due to the extraction method, since the selected temperature of the ultrasonic water bath was 65 °C, and according to the results obtained in the response surface model, the maximum extraction of phenolic compounds occurred at a water bath temperature of 50 °C for the mushroom *C. espinosae*, which, as already mentioned, are the compounds on which the antioxidant activity depends [50]. Regarding ergothioneine, its presence was evaluated in the aqueous extract obtained from the optimized extraction variables for the total soluble phenolic content, benefiting from the solubility of the molecule [53], and the antioxidant potential of both ergothioneine and total soluble phenolic content [54]. The non-detection of the of ergothioneine in the extract is consistent with a previous study on the digüeñe mushroom [55].

## 4. Conclusions

Drying is an appropriate method for preserving edible wild mushrooms, which are highly perishable due to their high moisture content. They have a short fruiting season, limiting the consumption area to the collection zone. In this study, it was observed that the different drying treatments applied significantly affected the physicochemical characteristics and antioxidant properties of the *C. espinosae* mushroom. The physicochemical properties of dried *C. espinosae* were evaluated through SEM images, rehydration rate and color parameters. The effects of drying treatments on structure and porosity depended on the drying process conditions, as observed in SEM images and rehydration rates. Mushrooms dried by microwave–vacuum at 210 W/20 kPa showed preserved structure and porosity comparable to freeze-dried mushrooms, which is accepted as a high-quality drying treatment for food matrices. This is consistent with the better rehydration rate observed alongside microwave–vacuum drying, with the rehydration mechanism following Fick’s diffusion law. For color parameters, the smallest difference in ΔE value was observed in freeze-dried mushrooms, followed by those dried by microwave-vacuum, again showing a value close to fresh mushrooms for the samples dried by microwave–vacuum at 210 W/20 kPa. Antioxidant properties of aqueous extracts from fresh and dried *C. espinosae* were evaluated through total soluble phenolic content, antioxidant activity and ergothioneine content, given the current interest in the antioxidant potential of edible wild mushrooms. Again, *C. espinosae* mushrooms dried by microwave–vacuum at 210 W/20 kPa showed better retention of total soluble phenolic content and antioxidant activity in the tested aqueous extracts, comparable to aqueous extracts of mushrooms dried by microwave–vacuum at 260 W/20 kPa; however, the physicochemical characteristics observed through SEM images and rehydration rates led to dismissing this latter option as an ideal alternative. Extraction methods influence the extraction capacity of the food matrix, and ultrasound-assisted extraction is a more promising technology for extracting bioactive compounds, used to maximize extraction capacity. Among the drying treatments applied to *C. espinosae* mushrooms, microwave–vacuum drying produced dried mushrooms with better overall physicochemical properties and antioxidant properties compared to *C. espinosae* mushrooms dried by freeze-drying and hot-air. Further research on edible wild mushrooms is needed to understand the influence of drying processes on the matrix and the extraction methods for the bioactive compounds found in dried mushrooms, especially for the extraction of phenolic compounds with interesting antioxidant activity.

## Figures and Tables

**Figure 1 jof-11-00013-f001:**
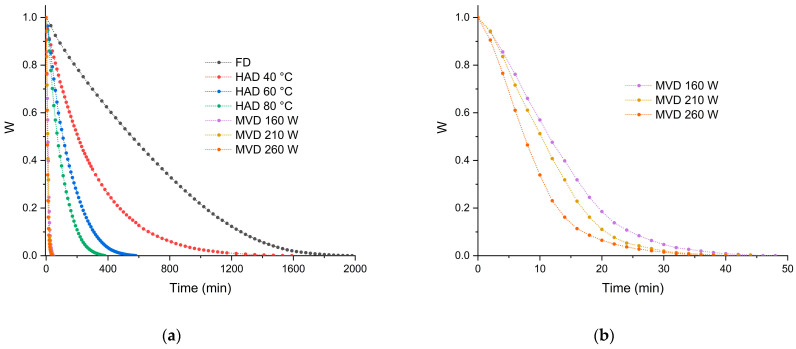
Moisture content curves of *C. espinosae* mushrooms dried by (**a**) different drying processes (FD: freeze-drying, HAD: hot-air drying and MVD: microwave–vacuum drying); and in detail by (**b**) microwave–vacuum at a vacuum pressure of 20 kPa for the three microwave powers.

**Figure 2 jof-11-00013-f002:**
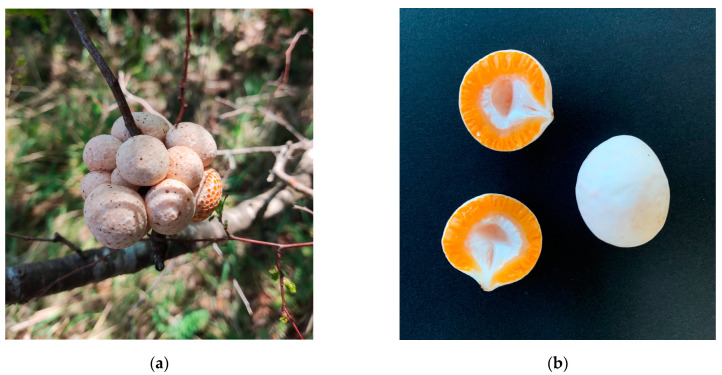
Macroscopic photos of fruiting bodies of the *C. espinosae* mushroom; (**a**) fruiting bodies in *Nothofagus obliqua* Forests (photographed by M.F.G.F.), (**b**) fruiting bodies before drying (photographed by the author), and (**c**) after drying, from left to right, freeze-dried (FD), hot-air dried (HAD) at 40, 60, and 80 °C and microwave–vacuum dried (MVD) at 160, 210, and 260 W at a vacuum pressure of 20 kPa.

**Figure 3 jof-11-00013-f003:**
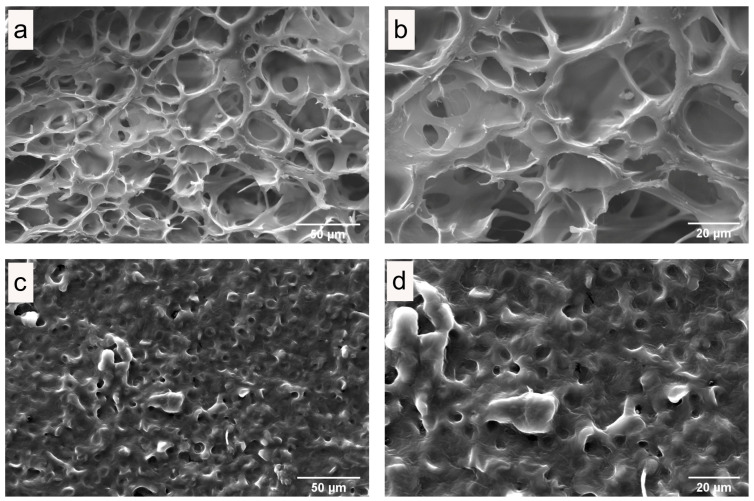
(**a**,**b**) Internal structure and (**c**,**d**) surface of freeze-dried *C. espinosae* by SEM. At magnification (**a**,**b**) 1000× and (**c**,**d**) 2000×.

**Figure 4 jof-11-00013-f004:**
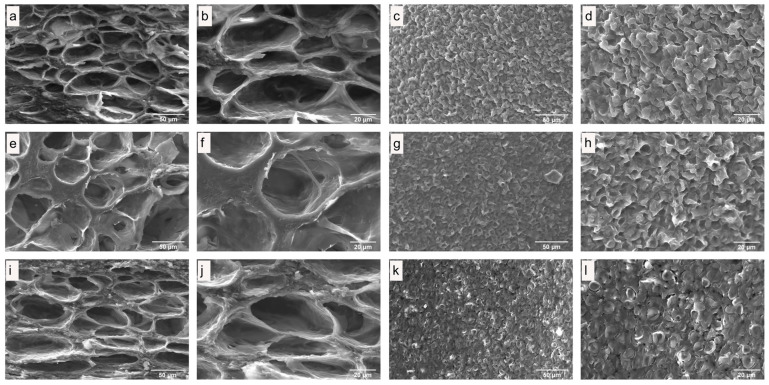
SEM images of the internal structure of *C. espinosae* dried by hot-air at temperatures of (**a**,**b**) 40 °C, (**e**,**f**) 60 °C, and (**i**,**j**) 80 °C. SEM images of the surface of the *C. espinosae* dried by hot-air at temperatures of (**c**,**d**) 40 °C, (**g**,**h**) 60 °C, and (**k**,**l**) 80 °C, and magnified (**a**,c,**e**,**g**,i,**k**) by 1000× and (**b**,**d,f,h,j**,**l**) 2000×.

**Figure 5 jof-11-00013-f005:**
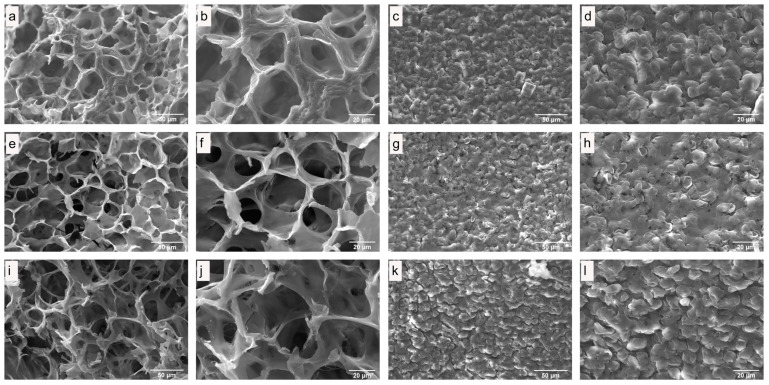
SEM images of the internal structure of the *C. espinosae* dried by microwave–vacuum at microwave power of (**a**,**b**) 160 W, (**e**,**f**) 210 W, and (**i**,**j**) 260 W at a vacuum pressure of 20 kPa. SEM images of the surface of the *C. espinosae* dried by microwave–vacuum at microwave power of (**c**,**d**) 160 W, (**g**,**h**) 210 W, and (**k**,**l**) 260 W at a vacuum pressure of 20 kPa, and magnified (**a**,**c**,**e**,**g**,i,**k**) by 1000× and (**b**,**d,f,h,j**,**l**) 2000×.

**Figure 6 jof-11-00013-f006:**
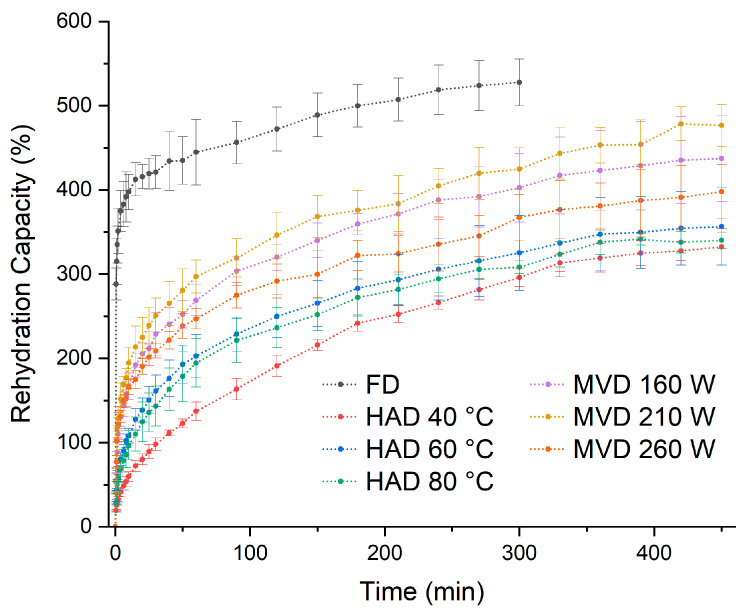
Rehydration rate curves of *C. espinosae* mushrooms dried by freeze-drying (FD), hot-air (HAD), and microwave–vacuum (MVD) at different times.

**Figure 7 jof-11-00013-f007:**
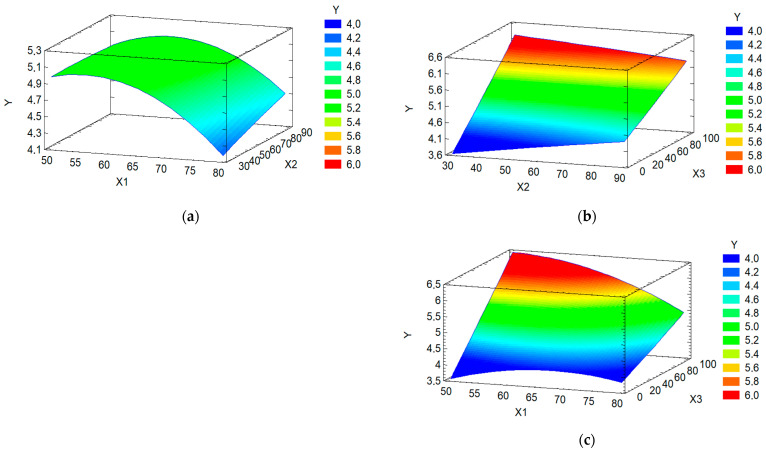
Response surface plots for total soluble phenolic content (Y) (mg GAE/g d.m.) of the factors and their interactions at (**a**) temperature (X_1_) (°C) and time (X_2_) (min); (**b**) time (X_2_) (min) and solvent (X_3_) (%); and (**c**) temperature (X_1_) (°C) and solvent (X_3_) (%).

**Figure 8 jof-11-00013-f008:**
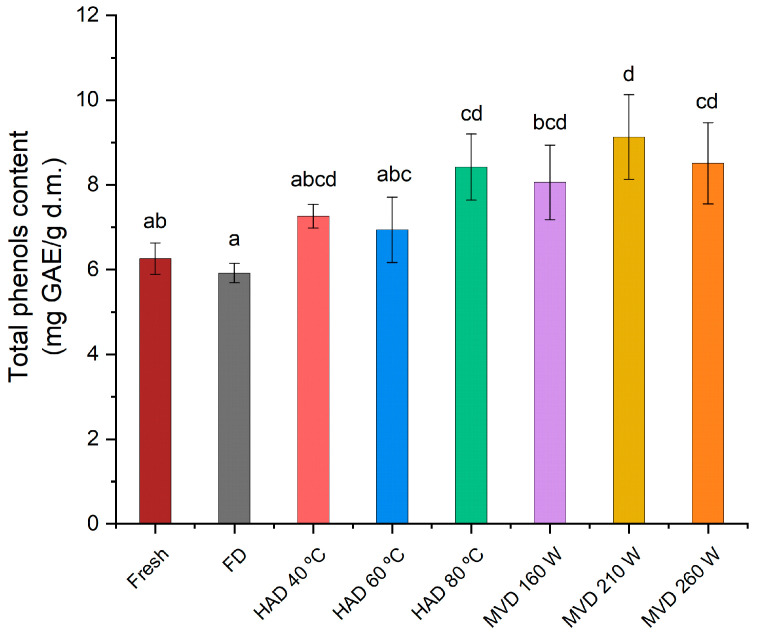
Total soluble phenolic content (mg GAE/g d.m.) of the aqueous extracts of fresh *C. espinosae* mushroom and dried by freeze-drying (FD), hot-air (HAD) and microwave–vacuum (MVD). The results were expressed in dry mass (d.m.). Different letters indicate a significant difference (*p* ≤ 0.05), according to Tukey’s test.

**Figure 9 jof-11-00013-f009:**
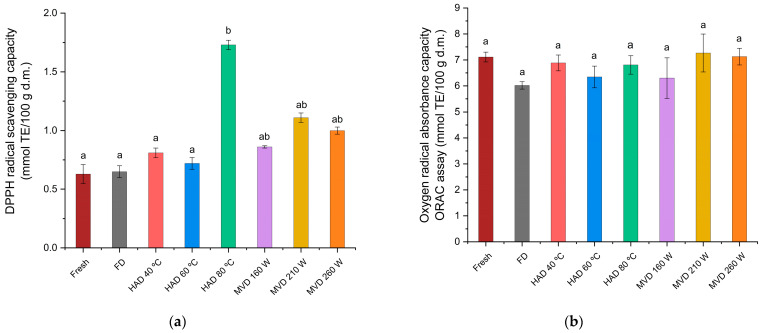
Antioxidant activity of aqueous extracts from fresh and dried *C. espinosae* mushrooms through (**a**) DPPH radical capture capacity (mmol TE/100 g d.m.) and (**b**) Oxygen radical absorption capacity (ORAC) (mmol TE/100 g d.m.) assays. The results are expressed in dry mass (d.m.). Different letters indicate a significant difference (*p* ≤ 0.05), according to Tukey’s test.

**Table 1 jof-11-00013-t001:** Kinetics models for the swelling of hydrogels ($\alpha =\frac{S_{t}}{S_{e}}$).

Kinetics Model	Equation	Reference
Peppas	$\alpha =kt^{n}$	[28,29]
One-dimensional diffusion	$\alpha^{2}=kt$	[28,29,30]
Two-dimensional diffusion	$\left(1-\alpha \right)ln\left(1-\alpha \right)+\alpha =kt$	[28,29,30]
Three-dimensional diffusion	[1−(1−α)13]2=kt	[28,29,30]

Where S_t_ and S_e_ are the absorbed mass at time t and at equilibrium respectively, k is the characteristic constant related to the lattice structure of the mushrooms and n is the diffusion exponent.

**Table 2 jof-11-00013-t002:** Fresh and dried *C. espinosae* mushrooms and their color parameters.

Conservation Status	L*	a*	b*	ΔE
Fresh	22.50 ± 0.86 ^d^	6.01 ± 0.03 ^bc^	15.58 ± 0.39 ^d^	28.02 ± 0.94 ^e^
FD	24.81 ± 0.38 ^e^	1.48 ± 0.06 ^a^	7.03 ± 0.21 ^a^	25.83 ± 0.32 ^d^
HAD 40 °C	11.47 ± 0.93 ^b^	5.35 ± 0.06 ^b^	10.20 ± 0.39 ^b^	16.26 ± 0.92 ^b^
HAD 60 °C	10.33 ± 0.31 ^b^	6.50 ± 0.17 ^c^	11.07 ± 0.43 ^b^	16.48 ± 0.55 ^b^
HAD 80 °C	8.81 ± 0.08 ^a^	5.21 ± 0.64 ^b^	8.16 ± 0.74 ^a^	13.09 ± 0.76 ^a^
MVD 160 W	17.90 ± 0.27 ^c^	6.33 ± 0.44 ^c^	13.35 ± 0.38 ^c^	23.22 ± 0.19 ^c^
MVD 210 W	17.73 ± 0.54 ^c^	5.96 ± 0.16 ^bc^	15.77 ± 0.51 ^d^	24.46 ± 0.72 ^cd^
MVD 260 W	18.09 ± 0.21 ^c^	6.82 ± 0.51 ^c^	14.17 ± 0.41 ^c^	23.97 ± 0.52 ^c^

Values are given as mean ± standard deviation of triplicate determinations. Different superscript letters indicate statistically significant differences between means (*p* < 0.05) for each parameter.

**Table 3 jof-11-00013-t003:** Kinetic models of hydrogel swelling for dried mushrooms.

Drying Processes	Peppas			One-Dimensional Diffusion		Two-Dimensional Diffusion		Three-dimensional Diffusion (Jander Equation)	
	k	n	R^2^	k	R^2^	k	R^2^	k	R^2^
FD	0.6256	0.14	99.77	0.1081	99.65	0.0827	99.79	0.0293	99.91
HAD 40 °C	0.0720	0.42	99.57	0.0028	99.95	0.0018	99.66	0.0005	98.77
HAD 60 °C	0.1358	0.36	99.44	0.0054	98.17	0.0035	99.11	0.0010	99.73
HAD 80 °C	0.1222	0.37	99.54	0.0052	99.60	0.0033	99.87	0.0010	99.72
MVD 160 W	0.1922	0.30	98.68	0.0063	93.10	0.0042	94.93	0.0012	96.63
MVD 210 W	0.2279	0.25	99.73	0.0063	93.89	0.0042	95.66	0.0013	97.30
MVD 260 W	0.2312	0.25	95.91	0.0068	92.53	0.0045	94.39	0.0013	96.12

**Table 4 jof-11-00013-t004:** Total soluble phenolic content of fresh *C. espinosae* mushrooms for the different combinations of extraction conditions based on the 3-level factorial experimental design in three blocks, with one central point per block, by Response Surface Methodology (RSM).

Run ^a^	Block	Temperature (X_1_) (°C)	Time (X_2_) (min)	Solvent (X_3_) (% ^b^)	Total Phenolic Content ^c^ (Y) (mg GAE/g d.m.)
1	1	65	60	0	3.60 ± 0.15
2	1	65	30	50	4.53 ± 0.64
3	1	50	90	50	5.02 ± 0.93
4	1	50	30	0	3.21 ± 0.61
5	1	80	90	0	4.29 ± 0.54
6	1	80	60	50	4.69 ± 0.67
7	1	80	30	100	4.76 ± 0.90
8	1	50	60	100	6.57 ± 0.38
9	1	65	90	100	6.11 ± 0.97
10	1	65	60	50	5.25 ± 0.36
11	2	65	90	50	4.83 ± 0.63
12	2	65	60	50	4.39 ± 0.49
13	2	80	60	0	3.70 ± 0.54
14	2	50	30	100	6.47 ± 1.13
15	2	50	60	50	5.55 ± 1.53
16	2	80	90	100	4.96 ± 0.95
17	2	50	90	0	3.77 ± 0.99
18	2	65	60	100	5.36 ± 0.36
19	2	65	30	0	3.77 ± 0.20
20	2	80	30	50	3.97 ± 0.17
21	3	65	90	0	4.59 ± 0.74
22	3	50	30	50	4.89 ± 0.73
23	3	65	60	50	5.58 ± 1.06
24	3	80	30	0	3.57 ± 0.77
25	3	80	60	100	5.03 ± 0.89
26	3	65	60	50	4.95 ± 0.41
27	3	65	30	100	7.13 ± 2.03
28	3	80	90	50	4.27 ± 0.27
29	3	50	90	100	5.91 ± 0.90
30	3	50	60	0	3.48 ± 0.85

^a^ Order of execution. ^b^ Percentage of distilled water in the solvent distilled water/ethanol. ^c^ Values are given as mean ± standard deviation of triplicate determinations; Total phenolic content expressed as mg GAE/g d.m.; d.m.: Dry mass.

**Table 5 jof-11-00013-t005:** One-way ANOVA of the independent variables (X_1_, X_2_, X_3_) for total soluble phenolic content.

Variables	*p*-Value
X_1_: Temperature (°C)	0.0050 *
X_2_: Time (min)	0.4207
X_3_: Solvent (%)	0.0000 *
X_1_^2^	0.0476
X_1_X_2_	0.4579
X_1_X_3_	0.0017 *
X_2_^2^	0.9338
X_2_X_3_	0.0262
X_3_^2^	0.8763

* Statistically significant differences (*p* < 0.05).

## Data Availability

The original contributions presented in the study are included in the article, further inquiries can be directed to the corresponding author.

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
