# Peer review of "Effects of Drying Treatments on the Physicochemical Characteristics and Antioxidant Properties of the Edible Wild Mushroom Cyttaria espinosae Lloyd (Digüeñe Mushroom)"

_jof, 2024, doi:10.3390/jof11010013_

Round 1

Reviewer 1 Report

 This work evaluated the influences of different drying methods on the physicochemical properties of Cyttaria espinosae, a kind of wild mushroom found in southern Chile. The wild mushrooms were subjected to freeze-drying, hot-air drying and microwave-vacuum drying, that were then assessed through rehydration capacity, color, microstructural properties by scanning electron microscopy (SEM), as well as total phenolic content, antioxidant activity determined by DPPH and ORAC assays, and ergothioneine levels of the fresh and dried C. espinosae extracts. The results showed that different drying treatments significantly affected the physical properties, total phenolic content and antioxidant activity. Among the three tested methods, microwave-vacuum drying was a suitable alternative for processing C. espinosae. These results have a practical meaning on the preservation processes of wild edible mushrooms.

A few points could be addressed before the manuscript can be considered for publication:

1) The abbreviation for “hot-air drying” can be HAD, instead of CD.

2) Lines 183 and 435: If the ratios of ethanol/distilled water were set at 100/0, 50/50, and 0/100% (v/v), the “Solvent (X3)” in Table 4 can be clearly defined as ethanol concentration (e.g. EtOH%).

3) Line 260: Please check the number “990”, it seems to be “590” in Figure 1.

4) Lines 272-274: Add notes for the abbreviations.

5) Line 347: Change “6e-f” into “6a-b”.

6) Lines 438- 440: Change “Table 3” into “Table 4”; change “Table 4” into “Table 5”.

7) Lines 457-478: I am puzzled if the X3 in Figure 10 (100% distilled water as solvent) is different from that in Table 4 and Table 5?

Also, Figure 10 should be reorganized again to show the missing numbers on the right sides.

8) Lines 552-554: Change “though” into “through”; change “(ORAC assay)(μmol ET/100 g d.m.)” into “(ORAC)(μmol ET/100 g d.m.) assays”.

This work evaluated the influences of different drying methods on the physicochemical properties of Cyttaria espinosae, a kind of wild mushroom found in southern Chile. The wild mushrooms were subjected to freeze-drying, hot-air drying and microwave-vacuum drying, that were then assessed through rehydration capacity, color, microstructural properties by scanning electron microscopy (SEM), as well as total phenolic content, antioxidant activity determined by DPPH and ORAC assays, and ergothioneine levels of the fresh and dried C. espinosae extracts. The results showed that different drying treatments significantly affected the physical properties, total phenolic content and antioxidant activity. Among the three tested methods, microwave-vacuum drying was a suitable alternative for processing C. espinosae. These results have a practical meaning on the preservation processes of wild edible mushrooms.

A few points could be addressed before the manuscript can be considered for publication:

1) The abbreviation for “hot-air drying” can be HAD, instead of CD.

2) Lines 183 and 435: If the ratios of ethanol/distilled water were set at 100/0, 50/50, and 0/100% (v/v), the “Solvent (X3)” in Table 4 can be clearly defined as ethanol concentration (e.g. EtOH%).

3) Line 260: Please check the number “990”, it seems to be “590” in Figure 1.

4) Lines 272-274: Add notes for the abbreviations.

5) Line 347: Change “6e-f” into “6a-b”.

6) Lines 438- 440: Change “Table 3” into “Table 4”; change “Table 4” into “Table 5”.

7) Lines 457-478: I am puzzled if the X3 in Figure 10 (100% distilled water as solvent) is different from that in Table 4 and Table 5?

Also, Figure 10 should be reorganized again to show the missing numbers on the right sides.

8) Lines 552-554: Change “though” into “through”; change “(ORAC assay)(μmol ET/100 g d.m.)” into “(ORAC)(μmol ET/100 g d.m.) assays”.

Reviewer 2 Report

1.The abstract does not highlight the main results, and the conclusion section is more like an abstract  

2.The length is too lengthy, especially in the results section, which should be integrated. For example, Figures 2-3 can be merged; Merge Figure 4-8 into two images; Figures 11-12 can also be merged  

3.Why did the conclusion come that 210 W/20 kPa tretment was best, there was no significant difference in total phenolic content and antioxidant capacity among different microwave treatments in figure11-12

4.Is there a result of the weight of the dried fruiting body. Generally speaking, edible mushrooms have a high moisture content and will experience significant weight loss after drying. In Figure 3, it appears that the weight of the sub entity is still relatively large after microwave treatment. So I have a question, shouldn't the active ingredients in fresh samples be lower than those in dry samples for the same one gram sample, because one gram of dry sample may be obtained from three grams of fresh sample, so its active ingredients are concentrated. But the results show that there seems to be little difference between the active ingredients

5.The samples were collected from 2022 and 2023, so how were the samples from different years handled in subsequent experiments?

1.The abstract does not highlight the main results, and the conclusion section is more like an abstract  

2.The length is too lengthy, especially in the results section, which should be integrated. For example, Figures 2-3 can be merged; Merge Figure 4-8 into two images; Figures 11-12 can also be merged  

3.Why did the conclusion come that 210 W/20 kPa tretment was best, there was no significant difference in total phenolic content and antioxidant capacity among different microwave treatments in figure11-12

4.Is there a result of the weight of the dried fruiting body. Generally speaking, edible mushrooms have a high moisture content and will experience significant weight loss after drying. In Figure 3, it appears that the weight of the sub entity is still relatively large after microwave treatment. So I have a question, shouldn't the active ingredients in fresh samples be lower than those in dry samples for the same one gram sample, because one gram of dry sample may be obtained from three grams of fresh sample, so its active ingredients are concentrated. But the results show that there seems to be little difference between the active ingredients

5.The samples were collected from 2022 and 2023, so how were the samples from different years handled in subsequent experiments?

Reviewer 3 Report

This study investigates the influence of three different drying methods on the physicochemical properties, total phenol content, ergothioneine content, and antioxidant activities of the wild edible mushroom Cittaria espinosae. In addition, the authors determined the conditions of variables such as temperature, time, and solvent through a response surface model to maximize the extraction of total phenol content. 

For C. espinosae, research on its bioactive compounds is still lacking. Its consumption is limited to certain areas and collection seasons (low availability for scientific research).

1. Line 174: Since the solvent is mentioned for the first time, it should be written which solvents were used for the extraction from fresh mushrooms (ethanol and/or water in the appropriate ratio) and from dried samples (water at 50 °C).

2. Lines 176, 205, 214: It is better to use g force than rpm because the size of the rotor can be different and the g force will be different while the rpm will remain the same (i.e. the larger the radius, the more acceleration is applied to the samples for the same rpm).

3. Lines 285-288, 290-291: The explanation of what the parameters L*, a*, b* and ΔE are is more appropriate for the Materials and Methods section than for the Results and Discussion.

4. Lines 549-550: It should be emphasized that ergothioneine was not detected in the extracts obtained by extraction with 100% distilled water at 50 °C for 30 minutes. These are the extraction conditions that are optimized for phenol extraction. In this study, optimization of ergothioneine extraction conditions was not performed. It should be stated (if available in scientific publications) what are the most optimal conditions for the extraction of ergothioneine.

5. It would be useful to present in the supplementary material the chromatograms recorded for the identification and quantification of ergothioneine.

6. The title of the paper refers only to the physicochemical properties. The antioxidant potential (biological activity) was also examined. The title should be adapted to all research objectives.

Round 2

Reviewer 1 Report

This work evaluated the influences of different drying methods on the physicochemical properties of Cyttaria espinosae, a kind of wild mushroom found in southern Chile. The wild mushrooms were subjected to freeze-drying, hot-air drying and microwave-vacuum drying, that were then assessed through rehydration capacity, color, microstructural properties by scanning electron microscopy (SEM), as well as total phenolic content, antioxidant activity determined by DPPH and ORAC assays, and ergothioneine levels of the fresh and dried C. espinosae extracts. The results showed that different drying treatments significantly affected the physical properties, total phenolic content and antioxidant activity. Among the three tested methods, microwave-vacuum drying was a suitable alternative for processing C. espinosae. These results have a practical meaning on the preservation processes of wild edible mushrooms. 

This work evaluated the influences of different drying methods on the physicochemical properties of Cyttaria espinosae, a kind of wild mushroom found in southern Chile. The wild mushrooms were subjected to freeze-drying, hot-air drying and microwave-vacuum drying, that were then assessed through rehydration capacity, color, microstructural properties by scanning electron microscopy (SEM), as well as total phenolic content, antioxidant activity determined by DPPH and ORAC assays, and ergothioneine levels of the fresh and dried C. espinosae extracts. The results showed that different drying treatments significantly affected the physical properties, total phenolic content and antioxidant activity. Among the three tested methods, microwave-vacuum drying was a suitable alternative for processing C. espinosae. These results have a practical meaning on the preservation processes of wild edible mushrooms. 

Author Response

Mr. Pakawat Sirilertpanich

E-Mail: sirilertpanich@mdpi.com

MDPI JoF Editorial Office

Manuscript: : jof-3355538

Dear Dr. Sirilertpanich

We are pleased to resubmit the revised version of our manuscript “Effects of drying treatments on the physicochemical characteristics and antioxidant properties of the edible wild mushroom Cyttaria espinosae Lloyd (Digüeñe mushroom).” for further consideration to publication in Journal of Fungi

In our response to reviewer’s comments, we described the changes introduced in the manuscript, and we hope to have met the requirement of the journal.

Sincerely yours,

Dr. Ociel Muñoz-Fariña

Reviewer 1

Major comments

This work evaluated the influences of different drying methods on the physicochemical properties of Cyttaria espinosae, a kind of wild mushroom found in southern Chile. The wild mushrooms were subjected to freeze-drying, hot-air drying and microwave-vacuum drying, that were then assessed through rehydration capacity, color, microstructural properties by scanning electron microscopy (SEM), as well as total phenolic content, antioxidant activity determined by DPPH and ORAC assays, and ergothioneine levels of the fresh and dried C. espinosae extracts. The results showed that different drying treatments significantly affected the physical properties, total phenolic content and antioxidant activity. Among the three tested methods, microwave-vacuum drying was a suitable alternative for processing C. espinosae. These results have a practical meaning on the preservation processes of wild edible mushrooms. 

Detail comments

This work evaluated the influences of different drying methods on the physicochemical properties of Cyttaria espinosae, a kind of wild mushroom found in southern Chile. The wild mushrooms were subjected to freeze-drying, hot-air drying and microwave-vacuum drying, that were then assessed through rehydration capacity, color, microstructural properties by scanning electron microscopy (SEM), as well as total phenolic content, antioxidant activity determined by DPPH and ORAC assays, and ergothioneine levels of the fresh and dried C. espinosae extracts. The results showed that different drying treatments significantly affected the physical properties, total phenolic content and antioxidant activity. Among the three tested methods, microwave-vacuum drying was a suitable alternative for processing C. espinosae. These results have a practical meaning on the preservation processes of wild edible mushrooms. 

Response

We sincerely appreciate the comments on our work titled "Evaluation of Drying Methods on the Physicochemical Properties of Cyttaria espinosae." Your observations are highly valuable and serve as a motivation to further deepen our analysis of preservation processes for wild mushrooms.

We are pleased that you highlighted the importance of evaluating the drying methods and the physicochemical properties of C. espinosae. As mentioned in the manuscript, our main objective was to compare three drying methods: freeze-drying, hot air drying, and vacuum microwave drying, using a combination of physical, chemical, and structural analyses. This approach enabled a comprehensive understanding of how these techniques influence the quality of the mushroom after processing.

It is true that vacuum microwave drying showed promising results, particularly in preserving antioxidant and phenolic properties as well as maintaining the physical characteristics of the mushroom. We believe this finding has significant practical implications for the food industry, especially in the context of wild mushrooms, where preserving their bioactive properties is crucial.

We fully agree that our results have relevant practical implications for the conservation and processing of edible mushrooms. Indeed, this study aims to serve as a foundation for future research focused on optimizing drying processes that can be applied at an industrial scale while maintaining the nutritional and functional values of mushrooms.

Once again, we express our gratitude for your valuable observations on our work. Your feedback reaffirms the significance of this study and inspires us to continue exploring innovative solutions for the preservation of wild mushrooms and other food products.

Additionally, we have conducted a thorough review of the manuscript and identified and corrected some details, as detailed below:

  1. The postal codes of the authors have been added.
  2. The names and affiliations of each author have been reviewed.
  3. Decimal punctuation in the numbers within tables and figures was corrected, replacing commas with periods.
  4. The figures were edited, adjusting the scales on the Y-axis (changing from commas to periods), and information was completed where necessary.
  5. "ET/100g" was replaced with "TE/100g" in the text and figures.
  6. Information about the equipment used was completed, including brand, model, city, and country.
  7. References were reviewed, correcting a few and ensuring there were no duplicates.
  8. The reference format was adjusted to conform to the MDPI format.

Reviewer 2 Report

No

It is suggested that the results total phenolic content and antioxidant capacity can also be merged.

Author Response

Mr. Pakawat Sirilertpanich

E-Mail: sirilertpanich@mdpi.com

MDPI JoF Editorial Office

Manuscript: : jof-3355538

Dear Dr. Sirilertpanich

We are pleased to resubmit the revised version of our manuscript “Effects of drying treatments on the physicochemical characteristics and antioxidant properties of the edible wild mushroom Cyttaria espinosae Lloyd (Digüeñe mushroom).” for further consideration to publication in Journal of Fungi

In our response to reviewer’s comments, we described the changes introduced in the manuscript, and we hope to have met the requirement of the journal.

Sincerely yours,

Dr. Ociel Muñoz-Fariña

Reviewer 2

Simplify the expression and it is best not to discuss it in the results section

  1. Section 3 of the manuscript is titled "Results and Discussion," as required by the Journal of Fungi formatting guidelines. Therefore, the results are discussed within this section. We believe this structure is appropriate, as it allows for a discussion of the findings as they are presented, minimizing redundancy. That said, we have reviewed the manuscript and will simplify the content where possible.

Major comments

No

Detail comments

It is suggested that the results total phenolic content and antioxidant capacity can also be merged.

  1. We appreciate the suggestion; however, we believe it is appropriate to present these items separately. This is because they involve distinct methodologies, have different interpretations, and address separate objectives.

Additionally, we have conducted a thorough review of the manuscript and identified and corrected some details, as detailed below:

  1. The postal codes of the authors have been added.
  2. The names and affiliations of each author have been reviewed.
  3. Decimal punctuation in the numbers within tables and figures was corrected, replacing commas with periods.
  4. The figures were edited, adjusting the scales on the Y-axis (changing from commas to periods), and information was completed where necessary.
  5. "ET/100g" was replaced with "TE/100g" in the text and figures.
  6. Information about the equipment used was completed, including brand, model, city, and country.
  7. References were reviewed, correcting a few and ensuring there were no duplicates.
  8. The reference format was adjusted to conform to the MDPI format.